# Advancing Healthcare in Low-Resource Environments Through an Optimization and Deployment Framework for Medical Multimodal Large Language Models

Aya El Mir*, Lukelo Thadei Luoga*, Boyuan Chen, Muhammad Abdullah Hanif, Muhammad Shafique

*eBRAIN Lab, Division of Engineering, New York University Abu Dhabi, UAE*

{ae2195, ltl2113, bc3194, mh6117, muhammad.shafique}@nyu.edu

*Abstract*—The critical shortage of medical professionals in low-resource countries, notably in Africa, hinders adequate healthcare delivery. AI, particularly Multimodal Large Language Models (MLLMs), can enhance the efficiency of healthcare systems by assisting in medical image analysis and diagnosis. However, the deployment of state-of-the-art MLLMs is limited in these regions due to the high computational demands that exceed the capabilities of consumer-grade GPUs. This paper presents a framework for optimizing MLLMs for resource-constrained environments. We introduce optimized medical MLLMs including TinyLLaVA-Med-F, a medical fine-tuned MLLM, and quantized variants (TinyLLaVA-Med-FQ4, TinyLLaVA-Med-FQ8, LLaVA-Med-Q4, and LLaVA-Med-Q8) that demonstrate substantial reductions in memory usage without significant loss in accuracy. Specifically, TinyLLaVA-Med-FQ4 achieves the greatest reductions, lowering dynamic memory by approximately 89% and static memory by 90% compared to LLaVA-Med. Similarly, LLaVA-Med-Q4 reduces dynamic memory by 65% and static memory by 67% compared to state-of-the-art LLaVA-Med. These memory reductions make these models feasible for deployment on consumer-grade GPUs such as RTX 3050. This research underscores the potential for deploying optimized MLLMs in low-resource settings, providing a foundation for future developments in accessible AI-driven healthcare solutions.

*Index Terms*—Artificial intelligence (AI), Clinical Applications, Medical Diagnostics, Memory Optimization, Multimodal Large Language Models (MLLMs), Quantization, Resource-Constrained Environments.

## I. INTRODUCTION

In many low-resource countries, a significant shortage of medical professionals critically hinders healthcare delivery. Figure 1 highlights that the ratio of medical doctors per 1,000 people in some regions is alarmingly low, which exacerbates the challenges in providing adequate healthcare services to these country's populations [1]. This issue is critical in Africa, especially in countries like Niger which has one of the world's lowest doctor-to-population ratios at only 0.03 doctors per 1,000 people, or one doctor for every 33,333 individuals. In contrast, countries like Canada had 2.46 doctors per 1,000 people as of 2021. This disparities calls for innovative solutions to bridge the gap between the limited number of healthcare providers and the increasing patient demands.

Artificial intelligence (AI) offers a promising solution to the challenges faced by healthcare systems, especially in countries with a low doctor-to-patient ratio. In these regions, doctors often overwork to serve many patients. While increasing the number of doctors seems like an obvious solution, many

*Equal contribution

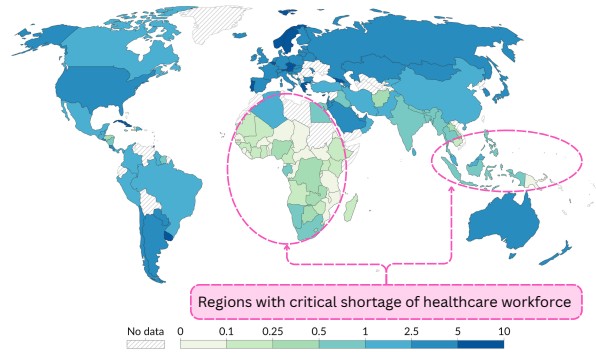

Fig. 1: Global distribution of medical doctors per 1,000 people in 2021 compiled by *the World Bank* and visualized by *Our World in Data* [1]. The map illustrates significant disparities in medical personnel availability, with particularly low ratios in many African countries.

countries cannot afford it due to high population levels and limited resources for medical education. Consequently, the need for doctors always surpasses the supply [2]. AI models can enhance the efficiency of limited doctors by consistently identifying subtle patterns and anomalies in medical images, reducing errors from human fatigue or oversight. [3]. This is crucial, as overburdened doctors might miss key abnormalities in diagnostic images such as CT scans or X-rays. Thus, AI can speed up the process of analyzing and interpreting medical images such as those used for radiology assessments, making healthcare delivery more efficient [4] [5]. Therefore, by integrating AI into healthcare systems in low-resource countries, we can improve the quality of care and ensure that even with a limited number of doctors, the healthcare system can meet the demands of the population. This advancement will lead to significant improvements in healthcare delivery, ultimately benefiting both doctors and patients.

Developed countries like Canada have harnessed AI to enhance healthcare delivery, improving efficiency, and patient outcomes. As a leader in AI integration, Canada benefits from a strong AI research base and significant historical contributions to AI. Examples include the Canadian Association of Radiologists, which provides guidelines on AI's impact on imaging practices, and Telus Health's Babylon app, which supports healthcare management and innovation. Additionally, Humber River Hospital in Toronto, North America's first fully digital hospital, leverages AI to boost patient care quality and healthcare delivery efficiency [6] [7].

As a subset of AI, Multi-modal Large Language Models (MLLMs) such as LLaVA-Med [8], Med-PaLM [9], Med-

Flamingo [10], PubMedCLIP [11], and BiomedCLIP [12], which integrate text and image modalities, are particularly impactful by enhancing image analysis [9]. MLLMs offer dialogue capabilities for open Visual Question Answering (VQA) settings, which are not present in traditional methods while showing effective competitive performance [13]. In healthcare, these models enhance few-shot learning, medical question answering, and conversational AI, demonstrating the potential of specialized MLLMs.

However, the deployment of MLLMs in low-resource countries is hindered by the lack of advanced computing resources. These regions often cannot afford the high-performance GPUs necessary for MLLMs, such as NVIDIA's A100 and V100, which are essential for managing the models' extensive computational demands. However, consumer-grade GPUs are more accessible and affordable in these settings. The issue lies in how state-of-the-art medical MLLMs, which are highly resource-hungry, cannot run on consumer-grade GPUs. These embedded devices have low computing power, storage capacity, and bandwidth, as they are designed primarily for applications like gaming and video editing [14]. Specifically, the autoregressive transformers used in MLLMs require substantial memory to process each text token sequentially, accessing large parameter sets that far exceed the limitations of consumer-grade GPUs [15]; i.e RTX 3050, RTX 3080, and Nvidia Jetson Orin. Consequently, low-resource countries are unable to benefit from the capabilities of medical MLLMs, which could otherwise help address their healthcare challenges, such as the low doctor-to-patient ratio. This constraint means that the regions most in need of MLLM's transformative potential in healthcare are the least likely to benefit from it [16]. To bridge this gap, MLLMs need to be optimized for consumer-grade hardware. By developing models that run effectively on less powerful GPUs, doctors in these regions can use MLLMs to assist in disease diagnosis, interpret medical images, and manage patient data more efficiently. This can enhance healthcare delivery, reduce the workload on the limited number of doctors, and improve patient outcomes in resource-constrained environments.

In exploring the current state of the art in MLLMs for medical applications, it is evident that there is a notable gap in the deployment of these models in resource-limited settings. One of the prominent models approaching this domain is Med-MoE (Mixture-of-Experts) [13]. Med-MoE, a lightweight framework optimized for resource-constrained environments, aligns medical images with language model tokens, performs task-specific instruction tuning, and includes domain-specific expert fine-tuning. With only 3.6 billion parameters, Med-MoE has demonstrated superior performance over LLaVA-Med [8] on multiple medical VQA datasets [13]. However, a significant limitation of Med-MoE is that it has yet to be deployed on a real resource-constrained device to prove its effectiveness in such environments, which is the main objective of its design. This leaves a significant gap in understanding its usability and efficiency in resource-constrained hospital settings. This issue underscores the need for more research to ensure that

medical MLLMs are not only designed for but also effectively deployed in resource-constrained devices, thus, making sure these models are accessible for low-resource countries.

Our paper aims to address this gap by making the following key novel contributions:

- **Framework for Optimizing MLLMs in Healthcare**: Our paper presents a framework for optimizing medical MLLMs by adapting the general-purpose TinyLLaVA [17] for the biomedical domain through extensive fine-tuning and quantization. Our framework introduces a family of optimized models: TinyLLaVA-Med-F (Fine-tuned), along with its quantized versions, TinyLLaVA-Med-FQ4 (4-bit) and TinyLLaVA-Med-FQ8 (8-bit). Additionally, we apply post-training quantization to LLaVA-Med, producing LLaVA-Med-Q4 (4-bit Quantized) and LLaVA-Med-Q8 (8-bit Quantized). These models not only meet or surpass the performance of existing state-of-the-art medical MLLMs but also demonstrate efficient memory consumption suitable for deployment on consumer-grade GPUs like RTX 3050, highlighting their potential for use in resource-limited settings.

- **Exploring the Tradeoff Between Performance and Memory Efficiency in Medical MLLMs**: Our study examines the tradeoff between performance and memory consumption in medical MLLMs. By integrating fine-tuning with optimization techniques like quantization and tuning hyperparameters, we create model variants that lie on the Pareto front, representing the optimal trade-offs between accuracy and memory consumption for a given memory constraint. This consideration of the Pareto front is key to developing models that maximize accuracy while adhering to specific resource limitations, such as the GPU memory available in portable or embedded devices. Our framework thus provides flexibility, offering model variants that can be adapted to different levels of GPU resources. This exploration lays the groundwork for future research on optimizing medical MLLMs, with an emphasis on balancing size reduction and maintaining medical accuracy.

- **Foundation for Future Research in Accessible MLLMs for Healthcare**: Our framework serves as a benchmark for future research, providing a blueprint for developing accessible MLLMs in healthcare and emphasizing the importance of deploying these models on consumer-grade GPUs. This approach also highlights practical usage scenarios, such as centralized deployment in hospitals where multiple doctors can benefit from the MLLM, rather than individual use on personal desktops. This deployment highlights the potential of our models to meet global healthcare demands, especially in resource-limited regions

## II. METHODOLOGY

Our methodology includes four key subsystems designed to effectively adapt MLLMs for resource-limited healthcare settings. The Optimization stage involves extensive fine-tuning of the general-purpose TinyLLaVA to adapt it for the biomedical domain, followed by post-training quantization of both TinyLLaVA-Med-F and LLaVA-Med to enhance their

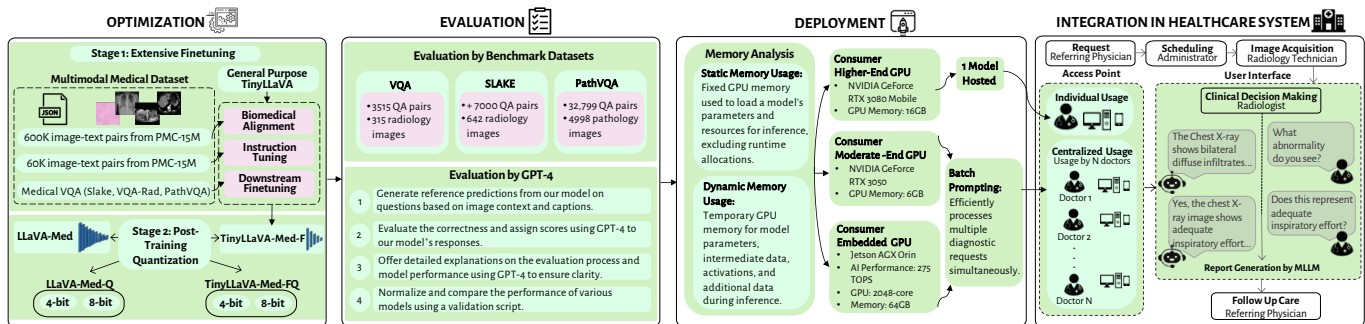

Fig. 2: Overview of the methodology framework across four key stages for adapting MLLMs to resource-limited healthcare settings. Starting with the Optimization phase, the general-purpose TinyLLaVA model undergoes fine-tuning and quantization into variants TinyLLaVA-Med-F, FQ4, FQ8 while the quantization of LLaVA-Med leads to variants LLaVA-Med-Q4 and Q8. The Evaluation phase tests the models on benchmark datasets (VQA, SLAKE, PathVQA) and with GPT-4. In the Deployment stage, models are implemented on consumer devices to assess memory usage. Finally, the Integration into Hospital Systems stage explores their integration into healthcare systems for improved radiology services.

efficiency and performance for medical applications. In the Evaluation phase, we rigorously test all these optimized models using benchmark datasets and comparative analyses to validate their performance. The Deployment stage focuses on deploying these models across consumer-grade GPU devices to study their memory consumption. Finally, in the Integration into Hospital Systems stage, we propose various ways these models can be integrated into healthcare systems, particularly to improve radiology services.

### A. Optimization

Our optimization process is designed to maintain high accuracy on medical domain queries while further reducing the inference cost and resource consumption of our models. This is achieved through Extensive Finetuning and Post-Training Quantization stages. These stages ensure that the models not only achieve state-of-the-art capabilities but also maintain low memory and computational requirements, making them suitable for deployment in consumer-grade GPU devices.

*1) Extensive Finetuning:* This stage involves extensive Finetuning of general-purpose TinyLLaVA [17] on a multimodal medical dataset comprising textual and image data. The TinyLLaVA model is already small by size as it only has 1.5B parameters, thus ensuring the final model of this optimization technique, TinyLLaVA-Med-F, will be smaller as compared to the state-of-the-art medical MLLMs. This optimization technique leverages the training stages of the LLaVA-Med model [8], a state-of-the-art multi-modal language model that has demonstrated high-performance metrics in medical domain applications. Unlike previous efforts to develop small-scale medical MLLMs such as TinyLLaVA-Med [18] — which did not utilize the full LLaVA-Med training pipeline, omitting the biomedical alignment stage, and consequently resulted in unsatisfactory accuracy — in this paper, we perform extensive finetuning that adapts the complete pipeline, including biomedical alignment. For the first two stages of fine-tuning (biomedical alignment and instruction tuning), we used the PMC-15M dataset [19], which contains 15 million high-quality biomedical image-text pairs from PubMed Central (PMC) publications. This dataset was chosen for its open-source accessibility and frequent use in state-of-the-art medical MLLMs [8] [12]. It provides diverse biomedical image types

offering broader coverage than earlier datasets like MIMIC-CXR, and addresses privacy concerns [20]. The following are the step-by-step processes of this optimization technique:

*a) Biomedical Alignment:* The first stage involves aligning the general-purpose TinyLLaVA model with biomedical contexts using 600,000 image-text pairs from the PMC-15M dataset [19]. These pairs are converted to instruction-following data, prompting the model to describe medical images accurately. During this stage, the visual encoder and language model weights remain frozen, focusing updates only on the projection matrix to align image features with biomedical text.

*b) Instruction Tuning:* The instruction-tuning process aims to adapt the model to follow detailed instructions, turning it into a conversational agent that can interact within biomedical contexts and perform tasks accurately. This stage relies on the Biomedical Instruction-Tuning Data from LLaVA-Med [8], sourced from PMC-15M [19], containing 60,000 image-text pairs across major imaging modalities. During the instruction-tuning stage, the visual encoder weights are kept frozen while the language model and projection layer weights are updated.

*c) Downstream Finetuning:* The final stage involves fine-tuning TinyLLaVA-Med on targeted biomedical VQA datasets; these datasets include VQA-RAD [21], SLAKE [22], and PathVQA [23]. VQA-RAD includes 3515 QA pairs across 315 radiology images, categorized into 11 types such as abnormality, size, and modality. SLAKE is a dataset with 642 radiology images and over 7000 QA pairs, while PathVQA offers 4998 pathology images with 32,799 QA pairs. The downstream finetuning is crucial for adapting the model to specific medical scenarios, enhancing its ability to respond to diagnostic questions with high accuracy.

Through the stages of Biomedical Alignment, Instruction Tuning, and Downstream Fine-tuning, we successfully transformed the general-purpose TinyLLaVA model into TinyLLaVA-Med-F.

*2) Post-training Quantization:* In this stage, we employ Post-Training Quantization (PTQ) to minimize the inference cost and resource consumption of TinyLLaVA-Med-F obtained from Stage 1. Additionally, we apply PTQ to LLaVA-Med, resulting in further optimized models. PTQ reduces computational costs by quantizing models post-training without requiring additional training [24]. The extensively fine-

tuned TinyLLaVA-Med-F model undergoes PTQ to produce 4-bit and 8-bit quantized versions; TinyLLaVA-Med-FQ4 and TinyLLaVA-Med-FQ8. These quantized versions offer even smaller, more efficient alternatives, making them particularly advantageous for low-memory environments. In addition to TinyLLaVA-Med-F, we applied PTQ to the LLaVA-Med model, a pre-trained 7-billion parameter model for biomedical applications. This effort aims to propose a method for obtaining a smaller MLLM without the need for high-end GPUs. Unlike the extensive fine-tuning needed to adapt TinyLLaVA for medical applications, PTQ of LLaVA-Med is less GPU-intensive as it bypasses the need for large training datasets [25]. Applying PTQ on LLaVA-Med using consumer-grade GPU laptops (i.e. portable computers with Intel Core i5 processors and RTX 3050 6GB GPUs) demonstrates that even state-of-the-art models can be efficiently quantized to 4-bit and 8-bit versions. This makes them accessible to those without the GPU resources to develop or fine-tune smaller general-purpose medical models. By including both LLaVA-Med-Q (LLaVA-Med-Q4 and LLaVA-Med-Q8) and TinyLLaVA-Med-FQ (TinyLLaVA-Med-FQ4 and TinyLLaVA-Med-FQ8) in all our results and evaluations, we are able to comprehensively assess the performance of these models.

### B. Evaluation

To ensure the robustness of our optimized models (LLaVA-Med-Q4, LLaVA-Med-Q8, TinyLLaVA-Med-F, TinyLLaVA-Med-FQ4, and TinyLLaVA-Med-FQ8) in medical contexts, we undertake a comprehensive evaluation. This involves two main approaches: benchmark dataset evaluation and comparative analysis using state-of-the-art models. More precisely, we assess the model's performance across various visual question-answering (VQA) datasets tailored for medical applications and leverage GPT-4 for comparative insights.

*1) Evaluation by Benchmark Datasets:* We assess the performance of our models using Visual Question-Answering (VQA) datasets tailored to medical contexts. These datasets include VQA-RAD [21], SLAKE [22] and PathVQA [23]. Regarding the performance metrics, we measure accuracy for closed-set questions and recall for open-set questions, ensuring a comprehensive evaluation of the model's capability to handle both straightforward and complex diagnostic queries.

*2) Evaluation by GPT-4:* To assess the performance of our models, including TinyLLaVA-Med-F, TinyLLaVA-Med-FQ4, TinyLLaVA-Med-FQ8, LLaVA-Med-Q4, and LLaVA-Med-Q8, we employ the process proposed by LLaVA-Med [8] to conduct a comparative analysis using state-of-the-art models like GPT-4 [26]. We use an evaluation dataset consisting of 193 novel questions based on 50 unseen image and caption pairs from PMC-15M [19]. The questions are of two types: conversational, derived from the same self-instruct data generation pipeline used in our model's second training stage, and detailed descriptions, randomly selected from a fixed set of queries designed to elicit comprehensive responses. We leverage GPT-4 to evaluate the correctness of our model's responses by comparing them against GPT-

4's reference predictions. GPT-4 assesses the responses from both our models and itself based on helpfulness, relevance, accuracy, and detail level, assigning an overall score from 1 (lowest) to 10 (highest). This comparative framework not only quantifies the performance of our optimized models but also provides insights into the models' usability in real-world applications through the detailed feedback from GPT-4.

### C. Deployment

The deployment of our MLLMs (TinyLLaVA-Med-F, TinyLLaVA-Med-FQ4, TinyLLaVA-Med-FQ8, LLaVA-Med-Q4, and LLaVA-Med-Q8) is implemented across a range of consumer-grade GPU devices, each selected based on its capability to meet different computational demands within hospital environments. Our deployment strategy includes devices such as the NVIDIA GeForce RTX 3050 with 6GB memory, which is mainly used for individual usage scenarios where a single doctor accesses the model. This setup is suitable for smaller clinics or remote settings where the diagnostic demands are manageable on a one-on-one basis due to the GPU's limited processing power and memory capacity. In contrast, consumer-grade GPU devices with higher computational power like the NVIDIA GeForce RTX 3080 with 16GB memory and the Jetson AGX Orin with 275 TOPS and 64GB of shared memory are utilized in settings with higher diagnostic demands. These higher-end GPUs support batch prompting techniques that allow multiple doctors to access a single MLLM simultaneously by batching diagnostic requests together, thereby optimizing throughput and minimizing latency [27] [28]. This is particularly advantageous in larger hospital settings where rapid and efficient processing of multiple simultaneous diagnostic requests is critical.

Additionally, detailed memory analysis is conducted to evaluate both static and dynamic memory usage during our model inference. This analysis is crucial for understanding how each model scales and adapts to different memory capacities, thus providing insights into the operational efficiency and viability of our models on resource-constrained embedded devices.

### D. Integration Strategies in Hospital Systems

This section outlines how our MLLMs could potentially be integrated into hospital systems, with a particular focus, for instance, on enhancing radiology services.

*1) Centralized Usage Potential:* Devices with significant computational capabilities, like the NVIDIA GeForce RTX 3080 and Jetson AGX Orin, can support batch prompting. This allows multiple doctors to access a single MLLM simultaneously, efficiently processing diagnostic requests in batches. Such centralized usage could significantly enhance throughput and reduce latency, ideally suited for busy hospital environments with high diagnostic demand.

*2) Individual Usage Potential:* In settings with limited computational resources, where only a Consumer Moderate-End GPU like the NVIDIA GeForce RTX 3050 could be available, individual usage of the MLLM may be more feasible. In these cases, the model would be hosted on the device

for exclusive use by a single doctor, thus avoiding the risks of system overload and ensuring stable performance. This deployment is targeting smaller clinics or remote areas, where diagnostic demands are manageable on an individual basis.

*3) Access Points and User Interface Design:* Whether through centralized or individual usage, all doctors would have the ability to interact with our models via their personal computing devices—desktops, laptops, or mobile phones. These access points connect to a user interface designed to facilitate the easy submission of diagnostic queries and the efficient reception of insights generated by the MLLM, ensuring a consistent and user-friendly experience across all user interactions.

*4) Radiological Integration Impact:* The typical radiology workflow begins with a request from the referring physician, followed by scheduling by an administrator, and image acquisition by a radiology technician, and integrates our models at the clinical decision-making stage. Radiologists can interact with the MLLM through a user interface on desktops, laptops, or phones. The model's conversational capability supports interactive queries allowing real-time clarification. This integration optimizes clinical workflows, facilitates faster decision-making, and improves patient outcomes through timely treatment. After using the MLLM, the radiologist can request a report, subsequently, the referring physician directs the patient to any necessary follow-up care.

## III. RESULTS

### A. Evaluation on Medical VQA datasets

The performance of our models (TinyLLaVA-Med-F, TinyLLaVA-Med-FQ4, TinyLLaVA-Med-FQ8, LLAVA-Med-Q8, LLAVA-Med-Q4) was assessed through benchmarks across multiple datasets including VQA-RAD, SLAKE, and PathVQA. The models were evaluated on their ability to answer both open and closed questions, comparing their effectiveness against other state-of-the-art models under supervised finetuning and zero-shot settings as seen in Table I. In supervised fine-tuning, models are explicitly trained on medical VQA datasets to optimize performance, whereas in the zero-shot setting, models are evaluated without any finetuning on medical VQA data. This demonstrates their out-of-the-box reasoning capabilities based on previously learned knowledge.

**Supervised Finetuning results:** TinyLLaVA-Med-F displayed high performance in most of the metrics for Medical VQA and even achieved higher accuracy scores than its counterparts like LLAVA-Med and TinyMoE-Med. This occurs in the case of open-ended questions in the SLAKE (85.43%) and PathVQA datasets (39.25%). These results showcase its effectiveness in deriving comprehensive answers from medical visuals and texts despite its compact size compared to state-of-the-art models.

**Zero-shot results:** TinyLLaVA-Med-F outperforms larger models such as LLAVA-Med and TinyMoE-Med in the closed-ended questions of VQA-RAD (68.01%). The TinyLLaVA-Med-FQ8 and TinyLLaVA-Med-FQ4 models, although they had lower accuracies in most datasets compared to larger models, showed minimal accuracy drops. This performance

TABLE I: Comparative Performance Analysis of Various Models on Medical Visual Question Answering Datasets. This table displays the accuracy percentages for both open and closed question types across three datasets: VQA-RAD, SLAKE, and PathVQA. It includes results from supervised finetuning experiments alongside zero-shot evaluations, highlighting the effectiveness of each model under different training conditions.

| Method | VQA-RAD | | SLAKE | | PathVQA | |
|---|---|---|---|---|---|---|
| | Open | Closed | Open | Closed | Open | Closed |
| TinyLLaVA-1.5B (Baseline) | 19.15 | 59.93 | 35.22 | 60.10 | 11.16 | 63.70 |
| **Our Supervised finetuning results (MLLM Based Methods)** | | | | | | |
| LLaVA | 50.00 | 65.07 | 78.18 | 63.22 | 7.74 | 63.2 |
| LLaVA-Med (LLama7B) | 61.52 | 84.19 | **85.34** | 85.34 | 37.95 | 91.21 |
| LLaVA-Med (Vicuna7B) | **64.39** | 81.98 | 84.71 | 83.17 | 38.87 | 91.65 |
| Med-Moe (Phi2:3.6B) | 58.55 | **82.72** | 85.06 | **85.58** | 34.74 | **91.98** |
| Med-Moe (StableLM:2.0B) | 50.08 | 80.07 | 83.16 | 83.41 | 33.79 | 91.30 |
| TinyLLaVA-Med-F (1.5B) | 50.6 | 81.25 | **85.34** | 85.43 | **39.25** | 90.56 |
| **Zero-shot results** | | | | | | |
| LLaVA-Med (LLama7B) | 36.23 | 60.16 | 41.72 | 47.6 | **10.86** | 59.75 |
| LLaVA-Med (Mistral7B) | **36.79** | 65.44 | 42.83 | 60.82 | 10.04 | 69.04 |
| LLaVA-Med-Q8 (Mistral7B) | 32.98 | **68.01** | 43.92 | **64.18** | 10.11 | **69.45** |
| LLaVA-Med-Q4 (Mistral7B) | 29.82 | 62.87 | **43.98** | 62.50 | 9.85 | 69.15 |
| Med-Moe (Phi2:3.6B) | 36.73 | 61.75 | 43.93 | 56.97 | 6.94 | 66.46 |
| Med-Moe (StableLM:2.0B) | 28.02 | 66.91 | 40.63 | 52.64 | 9.40 | 69.09 |
| TinyLLaVA-Med-F (1.5B) | 29.89 | **68.01** | 36.43 | 58.46 | 10.53 | 53.52 |
| TinyLLaVA-Med-FQ8 (1.5B) | 31.17 | 65.07 | 36.16 | 57.45 | 10.33 | 53.44 |
| TinyLLaVA-Med-FQ4 (1.5B) | 34.58 | 63.24 | 34.66 | 62.26 | 10.06 | 54.11 |
| **Representative non-MLLM Based SoTA methods (values from the literature)** | | | | | | |
| VL Encoder-Decoder [29] | 71.49 | 82.47 | | | 71.49 | 85.61 |
| Q2ATransformer [30] | 79.19 | 81.20 | | | 54.85 | 88.85 |
| Prefix T. Medical LM [31] | | | 84.30 | 82.01 | 40.00 | 87.00 |
| PubMedCLIP [11] | 60.10 | 80.00 | 78.40 | 82.50 | | |
| BiomedCLIP [12] | 67.60 | 79.80 | 82.05 | 89.70 | | |
| M2I2 [32] | 66.50 | 83.50 | 74.70 | 91.1 | 36.30 | 88.00 |

highlights that these smaller models can maintain relatively high accuracy levels, showcasing their potential in scenarios requiring efficient computational performance without significantly compromising accuracy, making them suitable for resource-constrained environments.

Furthermore, LLAVA-Med-Q8 showcased superior performance in the zero-shot setting, outperforming other models in closed-ended questions for SLAKE (64.18%) and VQA-RAD (68.01%), as well as in open-ended questions for PathVQA (69.45%). On the other hand, LLAVA-Med-Q4 excelled in open-ended questions for SLAKE (43.92%), demonstrating that even quantized MLLMs variants can achieve, and sometimes exceed, the accuracy of larger, state-of-the-art models.

These findings highlight the potential of our optimized models to deliver high accuracy in medical visual question-answering tasks, proving that size reduction does not necessarily compromise performance. This is especially crucial in resource-limited settings where deploying large-scale models may not be feasible.

Table I shows that both our models and state-of-the-art models perform poorly on the PathVQA open-ended questions, emphasizing the need for improvement in handling these questions. Overall, our models showed competitive results, highlighting the effectiveness of our optimization techniques and proving that our models are well-suited for medical applications, especially in low-resource settings where efficient processing and accurate diagnostics are critical.

### B. GPT-4 Evaluation:

This evaluation shows how each model handles conversational and descriptive question types within specific medical domains: Chest X-ray, MRI, Histology, Gross pathology, and CT Scan.

TABLE II: GPT-4 Evaluation of Models on Biomedical Multimodal Conversation. This table displays the performance across conversation and description question types, showing the models' proficiency in handling specific medical domains. The overall score reflects the average capability across all tested scenarios.

| Model | Conversation | Description | Chest X-Ray | MRI | Histology | Gross | CT Scan | Overall |
|---|---|---|---|---|---|---|---|---|
| TinyLLaVA (1.5B)-Baseline MLLM | 40.87 | 35.11 | 45.08 | 39.65 | 39.86 | 35.03 | 37.00 | 39.38 |
| LLaVA-Med (Mistral7b) | 59.57 | 52.59 | 64.04 | 48.82 | 63.68 | 54.31 | 56.89 | 57.77 |
| LLaVA-Med-Q8 (Mistral7b) | 60.03 | 50.23 | 61.71 | 48.52 | 63.21 | 58.20 | 55.22 | 57.49 |
| LLaVA-Med-Q4 (Mistral7b) | 58.65 | 48.94 | 61.00 | 47.96 | 63.31 | 53.36 | 53.88 | 56.14 |
| Med-Moe (Phi2:3.6B) | 55.49 | 43.79 | 60.37 | 46.68 | 55.91 | 47.11 | 51.40 | 52.46 |
| Med-Moe (StableLM:2.0B) | 52.99 | 40.81 | 56.44 | 44.29 | 54.03 | 50.37 | 43.91 | 49.83 |
| TinyLLaVA-Med-F (TinyLLaVA-1.5B) | 52.92 | 41.04 | 63.85 | 40.70 | 51.43 | 52.02 | 41.97 | 49.84 |
| TinyLLaVA-Med-FQ8 (TinyLLaVA-1.5B) | 53.80 | 39.89 | 63.13 | 42.09 | 54.96 | 46.55 | 43.80 | 50.20 |
| TinyLLaVA-Med-FQ4 (TinyLLaVA-1.5B) | 51.60 | 38.07 | 59.42 | 41.94 | 49.43 | 49.93 | 40.42 | 48.09 |

The GPT-4 evaluation results in Table II indicate that the overall scores of TinyLLaVA-Med-F, TinyLLaVA-Med-FQ8, TinyLLaVA-Med-FQ4, LLAVA-Med-Q8, and LLAVA-Med-Q4 are lower compared to state-of-the-art models. However, these models still deliver effective performance in medical conversations. Notably, LLAVA-Med-Q8 and LLAVA-Med-Q4 show robust capabilities across all domains, with LLAVA-Med-Q8 achieving an overall score of 57.49% and LLAVA-Med-Q4 achieving 56.14%, approaching the performance benchmarks set by leading models. On the other hand, the TinyLLaVA-Med-FQ8 achieved an overall score of 50.20% while TinyLLaVA-Med-FQ4 achieved a lower score of 48.09%.

Therefore, while our models have slightly lower scores, their performance is competitive with other models, effectively handling medical conversations. The GPT-4's evaluation, though not specialized for the medical domain, provides a reliable framework for assessing model capabilities by highlighting the conversational strengths and weaknesses of our models and specific areas for future enhancement.

### C. Memory Analysis Results

Memory efficiency is crucial for deploying models in resource-constrained environments. We conducted a comprehensive memory analysis, comparing both static and dynamic memory usage across various models, and correlated these metrics with model performance in selected VQA datasets accuracy and GPT-4 evaluation scores.

*1) Static and Dynamic Memory Consumption:* Figure 3 compares the memory consumption of our models with state-of-the-art models, highlighting reductions in both dynamic and static memory during inference.

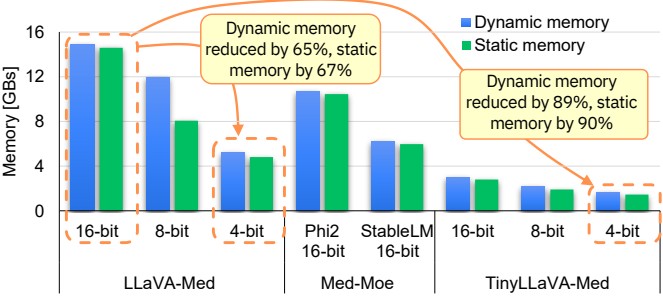

Fig. 3: Comparison of dynamic and static memory consumption across our models and other State-of-art MLLMs.

The TinyLLaVA-Med 4-bit version achieves the greatest reduction, lowering dynamic memory to 1.68 GB and static memory to 1.45 GB, an approximately 89% and 90% decrease respectively, compared to the 16-bit LLaVA-Med. Similarly, the 4-bit version of LLaVA-Med reduces dynamic memory to 5.18 GB and static memory to 4.81 GB, significantly less than its 16-bit counterpart's 14.86 GB and 14.59 GB.

These reductions demonstrate the potential of our optimization framework to substantially decrease the memory usage of these MLLMs, making them easy to deploy in GPU-constrained embedded devices.

*2) Memory Analysis by Model Accuracy Trade-off:* We analyzed the trade-off between dynamic memory consumption during zero-shot inference and the model's performance across different medical VQA datasets. Our approach included fitting a logarithmic line to the data, reflecting the principle that while larger models with more parameters initially capture more information, leading to improved performance, there is a point of diminishing returns where further increases in model size do not yield proportional gains in accuracy. Figure 4 (a) to (f) illustrate the memory-accuracy tradeoff for closed-ended and open-ended questions across different VQA datasets, of our models and state-of-art models.

*a) Closed-ended Questions Analysis:* For the VQA-RAD dataset specifically the close-ended questions (Figure 4 (a)), the TinyLLaVA-Med family models demonstrate a superior trade-off as they achieve higher accuracy while maintaining lower memory usage compared to other models. This trend is similarly observed in SLAKE (Figure 4 (b)). However, when it comes to the PathVQA dataset, (Figure 4 (c)) LLaVA-Med-Q4 model shows robust performance with significant memory efficiency, while the TinyLLaVA-Med family of models experiences a significant drop in accuracy.

*b) Open-ended Questions Analysis:* For the open-ended questions, the performance differs. In VQA-RAD (Figure 4 (d)), TinyLLaVA-Med-FQ8 and TinyLLaVA-Med-FQ4 show a promising tradeoff, outperforming other models, suggesting a higher utility for nuanced question types. For SLAKE (Figure 4 (e)), LLaVA-Med-Q4 maintains competitive accuracy with minimal memory use, reflecting its potential for resource-efficient applications. Lastly, for PathVQA (Figure 4 (f)), TinyLLaVA-MED models maintain low memory usage while achieving accuracies that challenge or surpass larger models.

Figure 4 highlights that our optimized variants of TinyLLaVA-Med and LLaVA-Med models (specifically the FQ4, FQ8, Q4) not only excel in managing the memory-

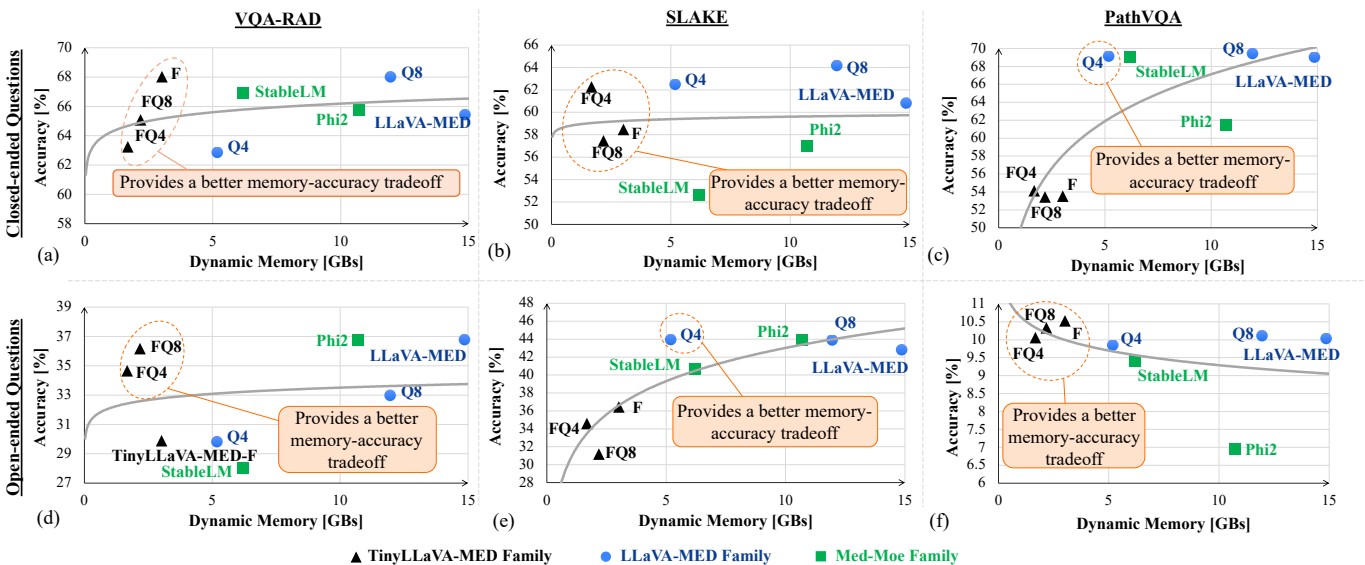

Fig. 4: Comparative analysis of memory-accuracy tradeoffs across three Visual Question Answering (VQA) datasets: VQA-RAD, SLAKE, and PathVOA. Notations are as follows: "FQ4" denotes a fine-tuned and quantized 4-bit version; "FQ8" refers to a fine-tuned and quantized 8-bit version; "Q4" signifies a quantized 4-bit version without fine-tuning; "Q8" indicates a quantized 8-bit version without fine-tuning. These plots demonstrate that our optimized models maintain accuracy with minimal memory usage.

accuracy tradeoff but also perform exceptionally well across various medical VQA domains. This indicates their suitability for deployment in medical settings where computational resources are limited but high accuracy is crucial.

## IV. DISCUSSION

**Evaluation and Benchmarking of Model Performance in the Healthcare field:** Our results show that our models excel at close-ended questions, demonstrating their potential for accuracy. However, their performance on open-ended questions, including many advanced models, is lacking, likely due to the complexities of healthcare. Therefore, we recognize that current evaluation benchmarks may not fully capture the models' ability to support doctors effectively, highlighting the need for further development to improve performance on real-world open-ended questions. This is a limitation noted in prior research [33]. Following the approach of other state-of-the-art models such as Med-Moe, we utilized established benchmarking datasets and methodologies introduced by LLaVA-Med [8], which include GPT-based evaluations to assess the conversational abilities of our models across different medical fields. These evaluation benchmarks provide insights into how the model's performance can be enhanced in specific areas such as radiology, histology, and others. In sum, we aimed for a comprehensive evaluation by comparing our models against state-of-the-art benchmarks, notably those used by LLaVA-Med. However, we acknowledge that more work is required to effectively assess the performance of Machine Learning (ML) models in healthcare. Future efforts should focus on validating clinical targets and metrics through clinical trials that measure tangible patient-relevant outcomes, such as reductions in mortality rate. [16].

**Paving the Path for ML in Healthcare:** Our work aims to encourage the development of Machine Learning

(ML) models that are specifically designed for healthcare rather than just applied to healthcare data. It is important to distinguish between "Machine Learning on healthcare data" and "Machine Learning for healthcare problems," recognizing that the technical novelty of the former does not inherently translate to impactful solutions for the healthcare domain [16]. Currently, most state-of-the-art datasets in healthcare come from developed countries, leading to an underrepresentation of low-resource regions where these technologies are most needed. As a result, MLLMs often fall short of addressing the specific healthcare challenges of these underserved areas. To ensure technology is more effective and widely adopted in these settings, it is essential to tailor training datasets to the specific needs and constraints of low-resource regions [16]. This can involve curating datasets with data from these areas or analyzing the geographical distribution of existing datasets, such as PMC-15M, to better align them with the healthcare realities in underserved communities.

State-of-the-art medical MLLMs are also developed mainly in high-income countries, yet the communities most in need, particularly in poorer regions, should be actively involved in their development [16]. This requires making the entire model-building pipeline, from training to deployment, accessible to these communities. By developing models that do not require substantial GPU resources, we are enabling individuals from these communities to further develop these models or build their own models, rather than relying on researchers from high-income countries. This approach ensures that the models are tailored to the specific capacities and needs of these regions, making the technology more likely to be adopted and effective [16]. By focusing on building capacity and involving low-resource countries in the innovation process, we can also address biases that stem from models developed in less diverse data environments. This inclusive approach, with a focus on impact, will lead to the creation of fairer and

more relevant models. Indeed, even the most advanced models will not drive change without a fundamental shift in how we apply AI in healthcare—one that prioritizes impact over performance. In low-resource countries, the emphasis must be on integrating AI into healthcare systems in ways that are accessible, sustainable, and tailored to the specific needs of the communities they are designed to serve.

**Need for Comprehensive Healthcare Improvements:** MLLMs offer significant potential to enhance healthcare in low-resource countries with low doctor-to-patient ratios but cannot replace the need for more physicians [2]. Although these models have the potential to enhance doctors' workflow and efficiency, addressing the shortage of healthcare professionals requires more than just technological solutions—it calls for greater investment in healthcare systems. Despite the emphasis on strengthening primary healthcare, efforts to improve healthcare access and quality often fall short due to limited resources and inadequate infrastructure [2]. Overcoming these challenges will require collaborative, sustained efforts that go beyond financial investment alone.

## V. CONCLUSIONS

Our study demonstrates the potential of optimized MLLMs in enhancing healthcare delivery in low-resource settings through accessible AI diagnostics. By leveraging optimization techniques and robust evaluations, we showcase our models' capability to operate on consumer-grade and specialized embedded GPUs without significantly compromising accuracy. While our results are promising, further research is needed to validate clinical outcomes through real-world trials.

### ACKNOWLEDGMENT

This work was partially supported by the New York University Abu Dhabi (NYUAD) Center for Artificial Intelligence and Robotics (CAIR), funded by Tamkeen under the NYUAD Research Institute Award CG010. We also acknowledge the NYU IT High Performance Computing facility for providing the required GPU resources and services for the large-scale experiments in this research.

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
