# OpenReview forum: "Advancing Healthcare in Low-Resource Environments Through an Optimization and Deployment Framework for Medical Multimodal Large Language Models"
_IEEE.org/EMBS/BHI/2024/Conference — IEEE BHI'24_

### Official Review · Reviewer_Msm8 · 2024-07-29
**The paper demonstrates how optimized AI models can improve healthcare efficiency in low-resource settings, but overlooks issues like training quality, and real-world validation**

**Overall Rating:** 7
**Confidence:** 1

**Other Quality Metrics:**

Clarity of writing: Good
Clinical Significance: Fair
Methodological Novelty: Good
Experiments and Results: Fair

**Questions For The Authors:**

- How do you ensure the quality and representativeness of the training data used for optimizing the AI models, especially given the diverse healthcare needs of low-resource settings?
- What measures are in place to provide ongoing technical support and infrastructure maintenance for these AI systems in low-resource environments?

**Strengths:**

- Comprehensive Evaluation: The researchers rigorously tested the models using benchmark datasets, proving their reliability and accuracy.
- Accessibility: The models run on consumer-grade GPUs, making advanced AI accessible to low-resource communities.
- Flexible Deployment: The deployment strategies cater to various computational environments, supporting both centralized and individual usage.
- Holistic Approach: The paper highlights the need for more medical professionals and investment, balancing the role of AI in healthcare improvement.

**Summary Of The Paper:**

The paper discusses advancing healthcare in low-resource settings using optimized Multimodal Large Language Models (MLLMs). It highlights the severe shortage of medical professionals in these regions and proposes AI as a viable solution to enhance healthcare efficiency. The researchers optimized general-purpose AI models for the biomedical domain, ensuring they can run on consumer-grade GPUs. Extensive testing demonstrated that these AI models can consistently identify subtle medical anomalies, aiding doctors significantly. The paper emphasizes the accessibility of these models, allowing individuals in low-resource areas to build and use them effectively. Various deployment strategies are suggested, including centralized and individual usage, to adapt to different computational capabilities. The study concludes that while AI can improve doctor efficiency, it cannot replace the need for more medical professionals. Future research should focus on real-world trials to validate the clinical outcomes of these AI models.

**Weaknesses:**

- Integration Challenges: The paper does not fully explore the practical challenges of integrating these AI models into existing healthcare workflows and systems.
- Technical Support: The paper lacks a discussion on the availability of technical support and infrastructure needed to troubleshoot and maintain AI systems in low-resource settings.
- Cost Implications: The financial aspects of deploying and maintaining these AI systems in low-resource settings are not explored.

---

### Official Review · Reviewer_41zp · 2024-08-08
**Great potential of deploying AI models in resource-constrained healthcare systems**

**Overall Rating:** 7
**Confidence:** 4

**Other Quality Metrics:**

•	Clarity of writing: Good
•	Clinical significance: Great
•	Methodological novelty: Good
•	Experiments and results: Good

**Questions For The Authors:**

None

**Strengths:**

Overall, the paper presents an interesting research question and a promising approach. They develop smaller, more efficient versions of a medical MLLM (TinyLLaVA-Med-F and its quantized variants) that significantly reduce memory usage without compromising performance. These optimized models can be deployed on consumer-grade GPUs, making them suitable for low-resource settings. Addresses a significant clinical problema, with a innovative approach, and potential for impact.

**Summary Of The Paper:**

The paper addresses the challenge of deploying complex AI models, specifically Multimodal Large Language Models (MLLMs), in resource-constrained healthcare settings. Due to a shortage of medical professionals in many countries, particularly in Africa, there is a critical need for AI solutions to augment healthcare delivery. To address this, the paper introduces a framework for optimizing MLLMs. The research demonstrates the potential of deploying advanced AI models in resource-constrained healthcare systems, contributing to improved healthcare accessibility and quality.

**Weaknesses:**

Limited scope. While MLLMs hold great promise in medical image analysis and diagnosis, their high computational demands limit their deployment in resource-constrained environments.

---

### Official Review · Reviewer_b63H · 2024-08-13

**Overall Rating:** 6
**Confidence:** 4

**Other Quality Metrics:**

(a) Clarity of writing - Good
(b) Clinical Significance - Good
(c) Methodological Novelty - Fair
(d) Experiments and Results - Good

**Questions For The Authors:**

See weaknesses

**Strengths:**

The paper is very well-written. The authors have identified a crucial issue that even though AI offers a promising solution to challenges faced by healthcare systems, but with the current requirements of deploying these models, it will not benefit the low-resource countries, where it is required the most. The paper presents the idea with great clarity and citing suitable examples such as the doctor-to-patient ratio in the global scenario. The authors present thorough experiments on various benchmarks which is very helpful to evaluate the current state of these models.

**Summary Of The Paper:**

The authors present a framework for optimizing MLLMs for resource constrained environments. Particularly, they create quantized versions of SOTA model LLaVA-Med. They also fine tune Tiny-LLaVA to create TinyLLaVA-Med-F, and its quantized variants. The authors fine tune on multimodal medical datasets, perform instruction tuning, and downstream fine tuning to successfully transform the general purpose model into medically-aligned MLLM. Finally, the authors apply post-training quantization resulting in further optimized models quantized to 4-bit and 8-bit versions.

**Weaknesses:**

1. First of all, given that authors propose a framework for fine-tuning over the Tiny-LLaVA model, it should have been included as a baseline in Table 1 and Table 2. Then only, it is possible to judge whether fine tuning has really aligned the model's responses to biomedical questions.

2. Secondly, the authors do not provide any details about the post-training quantization. If it only involves applying the method proposed in [22], the paper has very limited technical innovation. I would suggest the authors describe their approach comprehensively to assess the benefits of the proposed framework.  For ex. the authors could have described how they modify the projection matrix to align image features with biomedical text in an effective manner. Do they use a specific transformation from image to word-embedding space?

3. I understand that the models proposed in the model achieve substantial reductions in static and dynamic memory. However, if the framework merely includes fine-tuning it on medical data and then applying post-training quantization, the overall paper seems like an engineering design rather than an innovative contribution to the field. I would suggest the authors describe their approach in much greater detail for the readers' understanding.

4. I would also suggest that authors provide other metrics such as the training efficiency as improvement in accuracy is incremental. For example, reducing the rank of the base model can further decrease the size of the model and can even potentially decrease the training time. I suggest that the authors should provide a detailed ablation study in this regard.

---

### Decision · Program_Chairs · 2024-09-23

Accept